# Genomic analysis of virulence factors and antimicrobial resistance of group B *Streptococcus* isolated from pregnant women in northeastern Mexico

**Gerardo del Carmen Palacios-Saucedo**[1☯], **Lydia Guadalupe Rivera-Morales**[2☯]*, **José Manuel Vázquez-Guillén**[2☯], **Amilcar Caballero-Trejo**[3], **Melissa Carolina Mellado-García**[2], **Aldo Sebastián Flores-Flores**[2], **José Alfredo González-Navarro**[2], **Celia Geovana Herrera-Rivera**[4], **Luis Ernesto Osuna-Rosales**[5], **Julio Antonio Hernández-González**[6], **Ricardo Vázquez-Juárez**[6], **Carolina Barrón-Enríquez**[2], **Ramón Valladares-Trujillo**[7], **Joaquín Dario Treviño-Baez**[3], **César Alejandro Alonso-Téllez**[4], **Luis Daniel Ramírez-Calvillo**[4], **Ricardo Martín Cerda-Flores**[8], **Rocío Ortiz-López**[5], **Miguel Ángel Rivera-Alvarado**[1], **Fortino Solórzano-Santos**[9], **Jorge Castro-Garza**[10], **Cristina Rodríguez-Padilla**[2]

**1** División de Investigación en Salud y División de Auxiliares de Diagnóstico, Unidad Médica de Alta Especialidad Hospital de Especialidades No. 25, Instituto Mexicano del Seguro Social, Monterrey, Nuevo León, México, **2** Laboratorio de Inmunología y Virología, Facultad de Ciencias Biológicas, Universidad Autónoma de Nuevo León, San Nicolás de los Garza, Nuevo León, México, **3** Departamento de Epidemiología y Dirección de Educación e Investigación, Unidad Médica de Alta Especialidad No. 23 Hospital de Ginecología y Obstetricia "Dr. Ignacio Morones Prieto", Instituto Mexicano del Seguro Social, Monterrey, Nuevo León, México, **4** Dirección General de Calidad y Educación en Salud, Secretaría de Salud, Alcaldía Miguel Hidalgo, Ciudad de México, México, **5** Unidad de Genómica, Centro de Investigación y Desarrollo en Ciencias de la Salud, Universidad Autónoma de Nuevo León, Monterrey, Nuevo León, México, **6** Laboratorio de Genómica y Bioinformática, Centro de Investigaciones Biológicas del Noroeste, La Paz, Baja California Sur, México, **7** Coordinación de Educación e Investigación en Salud, Hospital General de Zona No. 17, Instituto Mexicano del Seguro Social, Monterrey, Nuevo León, México, **8** Facultad de Enfermería, Universidad Autónoma de Nuevo León, Monterrey, Nuevo León, México, **9** Hospital de Pediatría, Centro Médico Nacional Siglo XXI, Instituto Mexicano del Seguro Social, Alcaldía Cuauhtémoc, Ciudad de México, México, **10** Laboratorio de Bioquímica y Genética de Microorganismos, Facultad de Ciencias Biológicas, Universidad Autónoma de Nuevo León, San Nicolás de los Garza, Nuevo León, México

☯ These authors contributed equally to this work.
* lydiariver@gmail.com

**Data Availability Statement:** The laboratory protocols used have been deposited in protocols.io

## Abstract

## Introduction

Group B *Streptococcus* (GBS) causes infections in women during pregnancy and puerperium and invasive infections in newborns. The genes *lmb*, *cylE*, *scpB*, and *hvgA* are involved with increased virulence of GBS, and hypervirulent clones have been identified in different regions. In addition, increasing resistance of GBS to macrolides and lincosamides has been reported, so knowing the patterns of antibiotic resistance may be necessary to prevent and treat GBS infections. This study aimed to identify virulence genes and antibiotic resistance associated with GBS colonization in pregnant women from northeastern Mexico.

with accession numbers provided within the manuscript. In addition, all data obtained from whole-genome sequencing are deposited in the BioProject and BioSample databases at NCBI with the accession number PRJNA551699.

**Funding:** This study was supported by the Fondo de Investigación en Salud del Instituto Mexicano del Seguro Social (IMSS) (grant FIS/IMSS/PROT/PRIO/15/047, 2015). GCPS is supported by the Research Excellence Scholarship (Beca de Excelencia en Investigación) provided by the IMSS Foundation (Fundación IMSS, A.C.). The funders had no role in study design, data collection and analysis, decision to publish, or preparation of the manuscript.

**Competing interests:** The authors have declared that no competing interests exist.

## Methods

Pregnant women with 35–37 weeks of gestation underwent recto-vaginal swabbing. One swab was inoculated into Todd-Hewitt broth supplemented with gentamicin and nalidixic acid, a second swab was inoculated into LIM enrichment broth, and a third swab was submerged into a transport medium. All samples were subcultured onto blood agar. After overnight incubation, suggestive colonies with or without hemolysis were analyzed to confirm GBS identification by Gram staining, catalase test, hippurate hydrolysis, CAMP test, and incubation in a chromogenic medium. We used latex agglutination to confirm and serotype GBS isolates. Antibiotic resistance patterns were assessed by Vitek 2 and disk diffusion. Periumbilical, rectal and nasopharyngeal swabs were collected from some newborns of colonized mothers. All colonized women and their newborns were followed up for three months to assess the development of disease attributable to GBS. Draft genomes of all GBS isolates were obtained by whole-genome sequencing. In addition, bioinformatic analysis to identify genes encoding capsular polysaccharides and virulence factors was performed using BRIG, while antibiotic resistance genes were identified using the CARD database.

## Results

We found 17 GBS colonized women out of 1154 pregnant women (1.47%). None of the six newborns sampled were colonized, and no complications due to GBS were detected in pregnant women or newborns. Three isolates were serotype I, 5 serotype II, 3 serotype III, 4 serotype IV, and 2 serotype V. Ten distinct virulence gene profiles were identified, being *scpB*, *lmb*, *fbsA*, *acp*, *PI-1*, *PI-2a*, *cylE* the most common (3/14, 21%). The virulence genes identified were *scpB*, *lmb*, *cylE*, *PI-1*, *fbsA*, *PI-2a*, *acp*, *fbsB*, *PI-2b*, and *hvgA*. We identified resistance to tetracycline in 65% (11/17) of the isolates, intermediate susceptibility to clindamycin in 41% (7/17), and reduced susceptibility to ampicillin in 23.5% (4/17). The *tetM* gene associated to tetracyclines resistance was found in 79% (11/14) and the *mel* and *mefA* genes associated to macrolides resistance in 7% (1/14).

## Conclusions

The low prevalence of colonization and the non-occurrence of mother-to-child transmission suggest that the intentional search for GBS colonization in this population is not justified. Our results also suggest that risk factors should guide the use of intrapartum antibiotic prophylaxis. The detection of strains with genes coding virulence factors means that clones with pathogenic potential circulates in this region. On the other hand, the identification of decreased susceptibility to antibiotics from different antimicrobial categories shows the importance of adequately knowing the resistance patterns to prevent and to treat GBS perinatal infection.

## Introduction

Group B *Streptococcus* (GBS; *Streptococcus agalactiae*) colonizes the human genitourinary and gastrointestinal tract and causes a variety of infectious processes in pregnant women, such as asymptomatic bacteriuria, urinary tract infection, bacteremia, pneumonia, meningitis,

endocarditis, sepsis, and various obstetric complications as spontaneous abortion, premature labor, chorioamnionitis, endometritis, stillbirth, and neonatal and maternal death. In addition, according to the onset of clinical manifestations, GBS infection in newborns (NB) can present with bacteremia, pneumonia, and sepsis as early-onset disease, late-onset disease, or late-late-onset disease [1, 2]. Up to 40% of pregnant women present GBS colonization, and 1 to 2% of NB may develop infection by this microorganism [1, 3]. The main risk factor in developing GBS invasive neonatal disease is woman's recto-vaginal colonization during childbirth [4].

GBS is considered an uncommon cause of perinatal infections in Mexico and other Latin American countries [5]. Unlike developed countries, in Mexico, neither the intentional search for GBS colonization in pregnant women nor the administration of intrapartum antibiotic prophylaxis (IAP) to prevent perinatal infection by this microorganism is an established practice [6]. However, a study in Mexico reported high exposure to GBS in reproductive age women (90%) [7]. GBS colonization of pregnant women is reported in up to 13% in Mexico and between 10 and 40% worldwide [1, 8].

Previous studies reported several factors making GBS more virulent and resistant to antibiotics. The most studied virulence factor is the capsular polysaccharide (Cps), which defines GBS serotypes (Ia, Ib, II-IX) and contributes to evade the immune system. However, other factors, such as laminin-binding protein (Lmb), fibrinogens (Fbs), hypervirulent adhesin (HvgA), and alpha-C protein (ACP), are associated with adherence and cell invasion [9–11]. In addition, an increasing GBS resistance to macrolides, lincosamides, and tetracyclines has been reported by several authors worldwide [12, 13]. Multidrug resistance has been increased all over the world that is considered a public health threat. Several recent investigations reported the emergence of multidrug-resistant bacterial pathogens from different origins, including humans, birds, cattle, and fish, that increase the need for routine application of antimicrobial susceptibility testing to detect the antibiotic of choice and the screening of the emerging MDR strains [13–17]. The aim of the present study was to explore the presence of virulence and antibiotic resistance genes in GBS associated with colonization in pregnant women in a population from northeastern Mexico.

## Material and methods

### Design, participants, and clinical sampling

A cross-sectional study and subsequently a longitudinal study were conducted from April 2017 to December 2018 including pregnant women with 35–37 weeks of gestation in prenatal care at the Hospital de Ginecología y Obstetricia No. 23 of the Instituto Mexicano del Seguro Social (IMSS). Written informed consent was obtained from all participants. The study was approved by the IMSS Scientific Research National Committee (R-2014-785-069). The sample collection was performed taking a recto-vaginal swab according to the recommendations of the Centers for Disease Control and Prevention (CDC) of the United States of America (USA). Sample collection procedures are available in protocols.io at dx.doi.org/10.17504/protocols.io.b4ytqxwn. In addition, we collected demographic data, including the socioeconomic status using the Graffar-Méndez scale, previously validated and used in the Mexican population [18]. We also register the number of pregnancies, gestational age, route of delivery, type of delivery (dystocic or eutocic), premature rupture of membranes, asphyxia at birth, and whether or not GBS colonized the NB.

### Isolation and presumptive identification

One swab was inoculated into Todd-Hewitt broth supplemented with gentamicin 8 µg/ml and nalidixic acid 15 µg/ml (THB, Becton-Dickinson, NJ, USA). A second swab was inoculated

into LIM enrichment broth (Todd-Hewitt broth with 1% yeast extract, 15 μg/ml nalidixic acid, and 10 μg/ml colistin, Becton-Dickinson, New Jersey, USA). Finally, we submerged the third swab into a transport medium (ESwab, Copan Diagnostics, Brescia, Italy). The first two swabs were incubated 18–24 hours at 37 ˚C in 5% $CO_2$, then those 2 tubes along with the swab inside the transport medium were subcultured onto Petri dishes with Columbia agar (OXOID LTD, Basingstoke, Hampshire, England) with 5% sheep erythrocytes and incubated under the same conditions. After overnight incubation, suggestive colonies with or without hemolysis were analyzed to confirm GBS identification [19, 20]. GBS isolation procedure used is available in protocols.io at dx.doi.org/10.17504/protocols.io.b4ytqxwn.

## GBS identification and serotyping

The criteria for GBS identification were a) Gram-positive coccus; b) negative catalase test; c) positive hippurate hydrolysis; d) CAMP test with wedge-shaped hemolysis in the presence of *Staphylococcus aureus* ATCC 25923; and e) growth with color change from white to orange pigment after 18–24 h incubation in the Strep B Carrot Broth® chromogenic medium (Hardy Diagnostics®, Santa María, CA, USA). Serotyping was performed by latex agglutination following the manufacturer's instructions (ImmuLex® Strep-B Latex, Statens Serum Institute, Copenhagen, Denmark). Confirmed isolates were stored at -80 ˚C [19, 21]. GBS identification and serotyping procedures are available in protocols.io at dx.doi.org/10.17504/protocols.io.b4ytqxwn, dx.doi.org/10.17504/protocols.io.b4ygqxtw, dx.doi.org/10.17504/protocols.io.b4yjqxun, dx.doi.org/10.17504/protocols.io.b4ynqxve, dx.doi.org/10.17504/protocols.io.b4ymqxu6, and dx.doi.org/10.17504/protocols.io.b4ysqxwe.

## Antimicrobial resistance profile of the recovered GBS isolates

The categories of antimicrobials tested were fluoroquinolones, glycopeptides, glycylcyclines, oxazolidinones, penicillins, tetracyclines, in addition to erythromycin and clindamycin [22]. Antibiotic susceptibility testing was performed in the VITEK 2 system (BioMérieux, Marcy-L'Etoile, France) with $1.5x10^7$ CFU/ml of each GBS isolate loaded into the AST-GP75 card containing ampicillin, levofloxacin, moxifloxacin, linezolid, vancomycin, tetracycline, and tigecycline, according to the manufacturer's instructions. In addition, susceptibility to erythromycin and clindamycin was evaluated by standardized agar diffusion tests using discs with these antibiotics (Titan Biotech, Bhiwadi, Alwar, Rajasthan). The results were assessed by measuring the diameter of the inhibition zone and interpreted according to the CLSI guideline [23].

## Genomic DNA extraction and whole-genome sequencing

GBS isolates were cultured overnight at 37 ˚C in 5% $CO_2$ in 3 ml of THB. Cellular pellets generated by centrifugation (6,000 g for 10 min at 10˚C) were incubated with 180 μL of 50 mg/ml lysozyme for 60 min at 37 ˚C. The isolation of genomic DNA (gDNA) was made using the QIAamp DNA Mini Kit (QIAGEN, Hilden, Germany). The concentration of gDNA was assessed using the Quant-iT PicoGreen dsDNA Assay Kit (Thermo Fisher Scientific, Waltham, MA, USA) on the Qubit 2.0 Fluorometer (Thermo Fisher Scientific, Waltham, MA, USA), and libraries were prepared with the Nextera DNA Flex Library Prep Kit (Illumina Inc. San Diego, CA, USA) following manufacturer's instructions. Pooled DNA library was deposited on the sequencing cartridge (MiSeq Reagen Kit V2, Micro Flow Cell 300 cycles; Illumina Inc. San Diego, CA, USA) and sequenced in the MiSeq (Illumina Inc. San Diego, CA, USA) instrument.

## Bioinformatic data processing

The quality of raw sequence data reads was validated using the 0.11.8 version of the FastQC program [24], and draft genomes were *de novo* assembled with the A5-miseq program [25]. The Quast program was used to evaluate quality details of the assemblies, such as contigs quantity and length [26]. Assemblies with poor quality parameters were subjected to a QC-Filtering pre-cleaning script. Genome annotation was made with the Prokka software [27], which identifies genes and their products by homology analysis in prokaryotic genomes. The final identification and mapping of the GBS genes coding for the capsular polysaccharide and the virulence factors FbsA, FbsB, Lmb, ACP, PI-1, PI-2a, PI-2b, HvgA, CylE, and ScpB were performed by blast homology analysis with BRIG software [28]. The identification of antibiotic resistance genes was performed by comparison using the CARD database [29].

## Statistical analysis

The descriptive analysis of quantitative variables was carried out using medians with ranges and that of categorical variables with absolute frequencies and percentages. Differences between women colonized and not colonized by GBS in categorical variables were measured by either the $X^2$ test or Fisher's exact test and quantitative variables by the Mann-Whitney U test. The odds ratio and its 95% confidence interval were also measured. A p-value $<0.05$ was considered statistically significant. The analysis was made using the IBM SPSS Statistics software (Version 20; SPSS Inc., Armonk, NY, USA).

## Ethics

The study was approved and supervised by the National Committee for Scientific Research (Registration number: R-2014-785-069) of the Instituto Mexicano del Seguro Social. All enrolled women provided written informed consent before inclusion in the study.

## Results

### Obstetric characteristics and GBS colonization

We first sought to determine the prevalence of GBS in the studied population, including 1,154 pregnant women. Seventeen women were colonized by GBS for a prevalence of 1.47%. Most of women were under 30 years of age (73%), 15 of the 17 colonized women belonged to this same age group (88%). Out of the total sample, 352 (30.9%) were primiparous; instead, 8 (47.1%) out of the 17 women colonized by GBS were also in their first pregnancy. We did not record the age of 16 non-colonized women and the number of pregnancies of 13 non-colonized women. Among the non-colonized women, 49 (4.3%) had obesity, 209 (18.4%) gestational diabetes, and 21 (1.8%) type 2 diabetes (Table 1).

Pregnancy in the 17 GBS colonized women was delivered at term. Nine of these deliveries were vaginal and eutocic. Of the eight abdominal deliveries, four were iterative cesarean section, two for lack of progress in labor, and two for non-reassuring fetal status. Only 6 of the 17 NB of colonized mothers were sampled. None of them was colonized by GBS, and during the three months follow-up, no infectious events attributable to GBS occurred. However, one NB (patient No. 840) had pneumonia associated with birth asphyxia, requiring hospitalization for 20 days without identifying the etiologic microorganism. This NB received ampicillin (150 mg/kg/day) plus amikacin (15 mg/kg/day) without improvement after ten days of treatment. Therapy changed to piperacillin plus tazobactam (150 mg/kg/day piperacillin component) with a favorable evolution. None of the GBS colonized mothers developed infectious complications during the three-month postpartum follow-up. Regarding the serotypes of the 17 GBS

**Table 1. Sociodemographic and obstetric characteristics of 1154 pregnant women classified according to the status of group B *Streptococcus* (GBS) colonization.**

| | | GBS Colonization | | | |
|---|---|---|---|---|---|
| | **Total (n = 1154)** | **Yes (n = 17)** | **No (n = 1137)** | **RM (IC95%)** | ***p*** |
| Age (years) | 0 27 (14–43) | 25 (19–37) | 0027 (14–43) | - | 0.113 |
| Age group | | | | | |
| ≤30 years | 0833 (73.2%) | 15 (88.2%) | 0818 (73.0%) | 2.77 (0.63–12.22) | 0.125 |
| >30 years | 0305 (26.8%) | 02 (11.8%) | 0303 (27.0%) | | |
| Socioeconomic class* | | | | | |
| Middle, middle/low, or low | 0138 (12.0%) | 04 (23.5%) | 0134 (11.8%) | 2.30 (0.74–7.16) | 0.135 |
| Other | 1016 (88.0%) | 13 (76.5%) | 1003 (88.2%) | | |
| No. de pregnancies | | | | | |
| 1 | 0352 (30.9%) | 008 (47.1%) | 0344 (30.6%) | 2.01 (0.77–5.26) | 0.118 |
| 2 or more | 0789 (69.1%) | 009 (52.9%) | 780 (69.4%) | | |
| Co-morbidities | | | | | |
| Obesity | 0051 (4.4%) | 002 (11.8%) | 0049 (4.3%) | 2.96 (0.65–13.30) | 0.171 |
| Type 2 diabetes | 0021 (1.8%) | 0- | 0021 (1.8%) | - | 0.730 |
| Gestational diabetes | 0211 (18.3%) | 002 (11.8%) | 0209 (18.4%) | 0.59 (0.13–2.60) | 0.373 |

Results are presented as median (range) or absolute frequency (percentage).

*According to the Graffar-Méndez scale [18].

isolates, 3 were serotype I, 5 were serotype II, 3 were serotype III, 4 were serotype IV, and 2 were serotype V (Table 2).

## Antimicrobial resistance profile

All GBS isolates were susceptible to levofloxacin, moxifloxacin, linezolid, vancomycin, and tigecycline. Otherwise, 11 isolates (65%) were resistant to tetracycline. Isolates of serotypes Ia, II, III, and V were resistant to tetracycline, whereas the isolates of serotypes Ib and IV were susceptible to this antibiotic. Only one isolate was sensitive to all the antibiotics tested (isolate 755 of serotype IV). Concerning clindamycin, seven (41%) of the isolates had intermediate sensitivity. We also identified four isolates with reduced susceptibility to beta-lactam antibiotics: serotype Ib, III, IV, and V. From these later isolates, the serotype Ib and IV also had intermediate susceptibility to clindamycin (Table 3).

The classification of our isolates was done according to the international expert proposal for interim standard definitions for antimicrobial acquired resistance by Magiorakos AP et al. [22], following the categories and definitions proposed for *Enterococcus spp*. Seven of our isolates were non-susceptible to one agent in two antimicrobial categories tested, nine isolates were non-susceptible to one agent in one antimicrobial category tested, and only one isolate was susceptible to all antimicrobial categories tested (Table 4). Finally, we had 11 (64.7%) non-susceptible isolates in the tetracyclines category, 4 (23.6%) in the penicillins category, 7 (41.2%) non-susceptible to clindamycin and one non-susceptible to erythromycin (Table 5).

## Identification of Cps, virulence and antibiotic resistance genes from whole-genome analysis of GBS isolates

After assembling the complete draft genomes of 14 of the 17 GBS isolates, we conducted a comparative genomic analysis to identify genes encoding Cps, virulence factors, and antibiotic resistance. All data obtained from whole-genome sequencing are deposited in the BioProject

**Table 2. Clinical and obstetric characteristics of 17 pregnant women colonized by group B *Streptococcus* and 6 sampled newborns.**

| Patient No. | Type of delivery | Birth asphyxia | GBS colonization of NB | GBS serotype |
|---|---|---|---|---|
| 18 | Eutocic | No | No | Ib |
| 73 | Eutocic | No | No | IV |
| 141 | Eutocic | No | No | V |
| 204 | LPL | No | - | III |
| 210 | LPL | No | No | III |
| 414 | Iterative | No | - | V |
| 423 | Iterative | No | - | Ia |
| 480 | Uncertain | Yes[a] | - | II |
| 610 | Eutocic | No | - | II |
| 685 | Eutocic | No | - | II |
| 688 | Eutocic | No | - | IV |
| 725 | Eutocic | No | - | III |
| 755 | Eutocic | No | - | IV |
| 833 | Uncertain | No | No | II |
| 840 | Iterative | Yes[b] | No | II |
| 865 | Iterative | No | - | IV |
| 1038 | Eutocic | No | - | Ib |

NB: Newborn; LPL: Caesarean section due to lack of progress in labor; Iterative: Iterative cesarean section; Uncertain: Cesarean section due to non-reassuring fetal status.

[a] Use of balanced general anesthesia during surgery, Apgar 6/7.

[b] The NB was hospitalized due to pneumonia associated with neonatal asphyxia without the identification of the etiologic microorganism.

The patient was discharged 20 days later after clinical improvement.

and BioSample databases at NCBI with the accession number PRJNA551699. The genes identified in each of the 14 GBS isolates analyzed are described in Table 6. The Cps loci coding for the sialic acid capsular polysaccharide identified in the 14 isolates were Ia, Ib, II, III, IV, and V. We identified two different Cps loci in two isolates: Cps II and III in isolate 204, and Cps II and IV in isolate 865 (Table 6 and Fig 1).

We identified ten distinct virulence gene profiles; the most common profile was *scpB*, *lmb*, *fbsA*, *acp*, *PI-1*, *PI-2a*, and *cylE* (3/14, 21%). The genes *scpB*, *lmb*, and *cylE* were found in the fourteen isolates evaluated, and the *acp* gene encoding the alpha C protein was found in 10 (71.4%) isolates. The locus for virulence factor PI-1 was second in frequency with 12 positive isolates. *PI-2a* and *fbsA* genes were identified in 11 isolates, while *bca* gene was found in 10, *fbsB* gene was found in six isolates, and the loci for factor PI-2b and *hvg* gene were present in five isolates. Moreover, except for isolates 755 and 833, at least two loci of the Pili virulence factor (PI-1, PI-2a, or PI-2b) were simultaneously found in the genome of each isolate evaluated. Thus, the combination PI-1 + PI-2a was identified in eight isolates, and the combination PI-1 + PI-2b was identified in two. Furthermore, the triple combination of the PI-1 + PI-2a + PI-2b loci was found in isolates 204 and 865. In these two isolates, we identified all the genes coding for virulence factors evaluated (Tables 6 and 7 and Fig 2).

The *mprF* gene confers resistance to cationic peptides that disrupt the cell membrane, and it was found in all the isolates. The *tetM* gene associated with resistance to tetracyclines was identified in 11 isolates. While the *mel* and *mefA* genes conferring resistance to macrolides was found only in isolate 204 (Table 6).

**Table 3. Level of antibiotic resistance and serotype of 17 group B *Streptococcus* (GBS) isolated from pregnant women.**

| | | AMP | | LEV | | MOX | | LIN | | VAN | | TET | | TIG | | ERY (Disk 15µg)* | | | CLY (Disk 2µg)* | | |
|---|---|---|---|---|---|---|---|---|---|---|---|---|---|---|---|---|---|---|---|---|---|
| **Range IMC µg ml$^{-1}$** | | ≤ 0.25 | ≥ 16 | ≤ 0.12 | ≥ 8 | ≤ 0.25 | ≥ 8 | ≤ 0.5 | ≥ 8 | ≤ 0.5 | ≥ 32 | ≤ 1 | ≥ 16 | ≤ 0.12 | ≥ 2 | ≥ 21 | I 16–20 | ≤ 15 | ≥ 19 | I 16–18 | ≤ 15 |
| **Isolate No.** | **Serotype** | | | | | | | | | | | | | | | | | | | | | |
| 18 | Ib | RS | | S | | S | | S | | S | | S | | S | | S | | | I | | |
| 73 | IV | RS | | S | | S | | S | | S | | S | | S | | S | | | I | | |
| 141 | V | S | | S | | S | | S | | S | | R | | S | | S | | | S | | |
| 204 | III | RS | | S | | S | | S | | S | | R | | S | | S | | | S | | |
| 210 | III | S | | S | | S | | S | | S | | R | | S | | S | | | S | | |
| 414 | V | RS | | S | | S | | S | | S | | R | | S | | S | | | S | | |
| 423 | Ia | S | | S | | S | | S | | S | | R | | S | | S | | | S | | |
| 480 | II | S | | S | | S | | S | | S | | R | | S | | S | | | S | | |
| 610 | II | S | | S | | S | | S | | S | | R | | S | | S | | | S | | |
| 685 | II | S | | S | | S | | S | | S | | R | | S | | S | | | I | | |
| 688 | IV | S | | S | | S | | S | | S | | S | | S | | S | | | I | | |
| 725 | III | S | | S | | S | | S | | S | | R | | S | | S | | | S | | |
| 755 | IV | S | | S | | S | | S | | S | | S | | S | | S | | | S | | |
| 833 | II | S | | S | | S | | S | | S | | R | | S | | S | | | I | | |
| 840 | II | S | | S | | S | | S | | S | | R | | S | | S | | | I | | |
| 865 | IV | S | | S | | S | | S | | S | | S | | S | | I | | | S | | |
| 1038 | Ib | S | | S | | S | | S | | S | | S | | S | | S | | | I | | |

S: Sensible, I: Intermediate, R: Resistant, RS: Reduced susceptibility, AMP: Ampicillin, LEV: Levofloxacin, MOX: Moxifloxacin, LIN: Linezolid, VAN: Vancomycin, TET: Tetracycline, TIG, Tigecycline, ERY: Erythromycin, CLY: Clindamycin.

*Interpretive categories and zone diameter breakpoints (nearest whole mm).

## Discussion

The prevalence of GBS colonization in developed countries ranges from 20% to 30% in pregnant women [31]. In the present study, the prevalence was 1.47% in 1,154 pregnant women from Northeastern Mexico. The highest prevalence of GBS colonization is registered in the Dominican Republic (43.5%) and the lowest in Fiji (2%) and Argentina (1.4%); the latter is similar to our results [32, 33]. A survey carried out in the late 1980s in central Mexico identified GBS cervicovaginal colonization in 13% of 340 pregnant women [34]. While Ocampo-Torres et al., found GBS colonization in 8.6% of pregnant women in a region of southeastern Mexico [35].

In the present study, most women were younger than 30 years, as also were 15 of the 17 GBS colonized women (88.2%). Several studies have demonstrated that children of young mothers (less than 20 years old) have a higher risk of developing GBS early-onset disease [36]. We did not find any newborn with early-onset disease or other complication due to GBS infection, no matter the age of the mother. We found that 23% (4) of the GBS colonized women belonged to a socioeconomic level of relative or critical poverty, but only 12% of the non-colonized women belonged to the same group. A study from Southeastern Mexico classified women according to the degree of marginalization, finding that pregnant women with high and very high socioeconomic marginalization had a prevalence of GBS colonization 1.7 times higher (12.7%) than the other groups [35]. Our results also show that GBS colonization was independent on the number of pregnancies, which contrasts to the report by Anthony et al. in California in 1978, where colonization rates were lower in women after the fourth pregnancy [37].

**Table 4. Antimicrobial susceptibility and antimicrobial categories of 17 group B *Streptococcus*.**

| Isolate No. | Serotype | Antimicrobials tested | | | | | | | | | No. of NS antimicrobials* | No. of NS categories* |
|---|---|---|---|---|---|---|---|---|---|---|---|---|
| | | AMP | LEV | MOX | LIN | VAN | TET | TIG | ERY | CLY | | |
| 18 | Ib | RS | S | S | S | S | S | S | S | I | 2 | 2 |
| 73 | IV | RS | S | S | S | S | S | S | S | I | 2 | 2 |
| 141 | V | S | S | S | S | S | R | S | S | S | 1 | 1 |
| 204 | III | RS | S | S | S | S | R | S | S | S | 2 | 2 |
| 210 | III | S | S | S | S | S | R | S | S | S | 1 | 1 |
| 414 | V | RS | S | S | S | S | R | S | S | S | 2 | 2 |
| 423 | Ia | S | S | S | S | S | R | S | S | S | 1 | 1 |
| 480 | II | S | S | S | S | S | R | S | S | S | 1 | 1 |
| 610 | II | S | S | S | S | S | R | S | S | S | 1 | 1 |
| 685 | II | S | S | S | S | S | R | S | S | I | 2 | 2 |
| 688 | IV | S | S | S | S | S | S | S | S | I | 1 | 1 |
| 725 | III | S | S | S | S | S | R | S | S | S | 1 | 1 |
| 755 | IV | S | S | S | S | S | S | S | S | S | 0 | 0 |
| 833 | II | S | S | S | S | S | R | S | S | I | 2 | 2 |
| 840 | II | S | S | S | S | S | R | S | S | I | 2 | 2 |
| 865 | IV | S | S | S | S | S | S | S | I | S | 1 | 1 |
| 1038 | Ib | S | S | S | S | S | S | S | S | I | 1 | 1 |

*Number of antimicrobials to which each isolate was non-susceptible (NS) according to the categories and definitions for *Enterococcus spp* by Magiorakos AP et al. [22].
S: Sensible, I: Intermediate, R: Resistant, RS: Reduced susceptibility, AMP: Ampicillin, LEV: Levofloxacin, MOX: Moxifloxacin, LIN: Linezolid, VAN: Vancomycin, TET: Tetracycline, TIG, Tigecycline, ERY: Erythromycin, CLY: Clindamycin.

A systematic review and meta-analysis reported that 48.9% cases of GBS invasive disease in children younger than three months old are caused by serotype III. Although the predominant colonizing serotypes around the world were Ia representing 22.9%, Ib 7.0%, II 6.2%, and V 9.1% [38]. In the USA and Europe, the serotype causing most cases of GBS invasive disease is serotype III [3]. In the present study six serotypes were identified, predominating serotype II, IV, and III. This finding contrasts with studies carried out in central Mexico in the 1980s, in which serotype I predominated (33%) with low participation of serotype III (3%) and a high prevalence of non-typeable isolates (18.2%) [34]. Subsequent studies reported type I as the predominant serotype in central and western Mexico (58.8–61.3%), with a greater number of

**Table 5. Antimicrobial resistance of 17 group B *Streptococcus* according to the categories and definitions for *Enterococcus spp* by Magiorakos et al. [22].**

| Antimicrobial categories | Antimicrobial agents | No. of non-susceptible GBS isolates |
|---|---|---|
| Fluoroquinolones | Levofloxacin | 0 |
| | Moxifloxacin | 0 |
| Glycopeptides | Vancomycin | 0 |
| Oxazolidinones | Linezolid | 0 |
| Penicillins | Ampicillin | 4 (23.6%) |
| Tetracyclines | Tetracycline | 11 (64.7%) |
| | Erythromycin | 1 (5.9%) |
| | Clindamycin | 7 (41.2%) |

GBS: group *B Streptococcus*.

**Table 6. Capsular polysaccharide (Cps), virulence factors, and antibiotic resistance genes in group B *Streptococcus* (GBS) isolated from pregnant women.**

| Virulence factor[a] | GBS isolate number | | | | | | | | | | | | | | Reference[b] |
|---|---|---|---|---|---|---|---|---|---|---|---|---|---|---|---|
| | 73 | 141 | 204 | 210 | 414 | 423 | 480 | 610 | 725 | 755 | 833 | 840 | 865 | 1038 | |
| **Immune evasion** | | | | | | | | | | | | | | | |
| Serotype[c] | IV | V | III | III | V | Ia | II | II | III | IV | II | II | IV | Ib | |
| Capsular polysaccharide [d] | | | | | | | | | | | | | | | |
| Cps Ia | - | - | - | - | - | + | - | - | - | - | - | - | - | - | LT671983.1 |
| Cps Ib | - | - | - | - | - | - | - | - | - | - | - | - | - | + | LT671984.1 |
| Cps II | - | - | + | - | - | - | + | + | - | - | + | + | + | - | LT671985.1 |
| Cps III | - | - | + | + | - | - | - | - | + | - | - | - | - | - | LT671986.1 |
| Cps IV | + | - | - | - | - | - | - | - | - | + | - | - | + | - | LT671987.1 |
| Cps V | - | + | - | - | + | - | - | - | - | - | - | - | - | - | LT671988.1 |
| Cps VI | - | - | - | - | - | - | - | - | - | - | - | - | - | - | LT671989.1 |
| Cps VII | - | - | - | - | - | - | - | - | - | - | - | - | - | - | LT671990.1 |
| Cps VIII | - | - | - | - | - | - | - | - | - | - | - | - | - | - | LT671991.1 |
| Cps IX | - | - | - | - | - | - | - | - | - | - | - | - | - | - | LT671992.1 |
| *scpB* | + | + | + | + | + | + | + | + | + | + | + | + | + | + | U56908.1 |
| **Cell adhesion and invasiveness** | | | | | | | | | | | | | | | |
| *lmb* | + | + | + | + | + | + | + | + | + | + | + | + | + | + | AF062533.1 |
| *fbsA* | + | - | + | + | - | + | + | - | + | + | + | + | + | + | AJ437620.1 |
| *fbsB* | - | - | + | + | - | + | - | - | + | + | - | - | + | - | HQ267707.1 |
| *acp* | - | - | + | + | + | + | + | - | - | + | + | + | + | + | M97256.1 |
| *hvgA* | - | - | + | + | - | - | - | - | + | + | - | - | + | - | CP020432.2 |
| **Antimicrobial peptide resistance** | | | | | | | | | | | | | | | |
| Pili | | | | | | | | | | | | | | | |
| PI-1 | + | + | + | + | + | + | + | + | + | - | - | + | + | + | EU929743.1 |
| PI-2a | + | + | + | - | + | + | + | + | - | - | + | + | + | + | EU929327.1 |
| PI-2b | - | - | + | + | - | - | - | - | + | + | - | - | + | - | EU929402.1 |
| **Pore-forming toxin** | | | | | | | | | | | | | | | |
| *cylE* | + | + | + | + | + | + | + | + | + | + | + | + | + | + | AF093787.2 |
| **Antibiotic resistance** | | | | | | | | | | | | | | | |
| Perfect identity[e] | | | | | | | | | | | | | | | |
| *mel* | - | - | + | - | - | - | - | - | - | - | - | - | - | - | |
| Strict identity[f] | | | | | | | | | | | | | | | |
| *mprF* | + | + | + | + | + | + | + | + | + | + | + | + | + | | |
| *tetM* | - | + | + | + | + | - | + | + | + | - | + | + | + | + | |
| *mefA* | - | - | + | - | - | - | - | - | - | - | - | - | - | - | |

[a] Classification obtained from Rajagopal L, 2009 [30].

[b] GenBank ID of the sequences used as a reference for gene identification.

[c] Determined by latex agglutination of the capsular polysaccharide-type antigen with the StrepPRO™ Streptococcal Grouping Kit (Hardy Diagnostics; Santa Maria, CA, USA).

[d] Locus Cps mapping by whole genome sequencing.

[e] Detection carried out as clinical surveillance finding 100% coincidences.

[f] Detection of previously reported variants in which mutations may exist finding probably functional genes (https://card.mcmaster.ca/).

serotype III isolates (5.9–12.8%) and a lower proportion of non-typeable isolates (0.0–5.9%) than previously reported [39, 40].

A decreasing in the frequency of early-onset neonatal GBS disease has been related to the implementation of screening programs and the use of IAP. A retrospective study in central

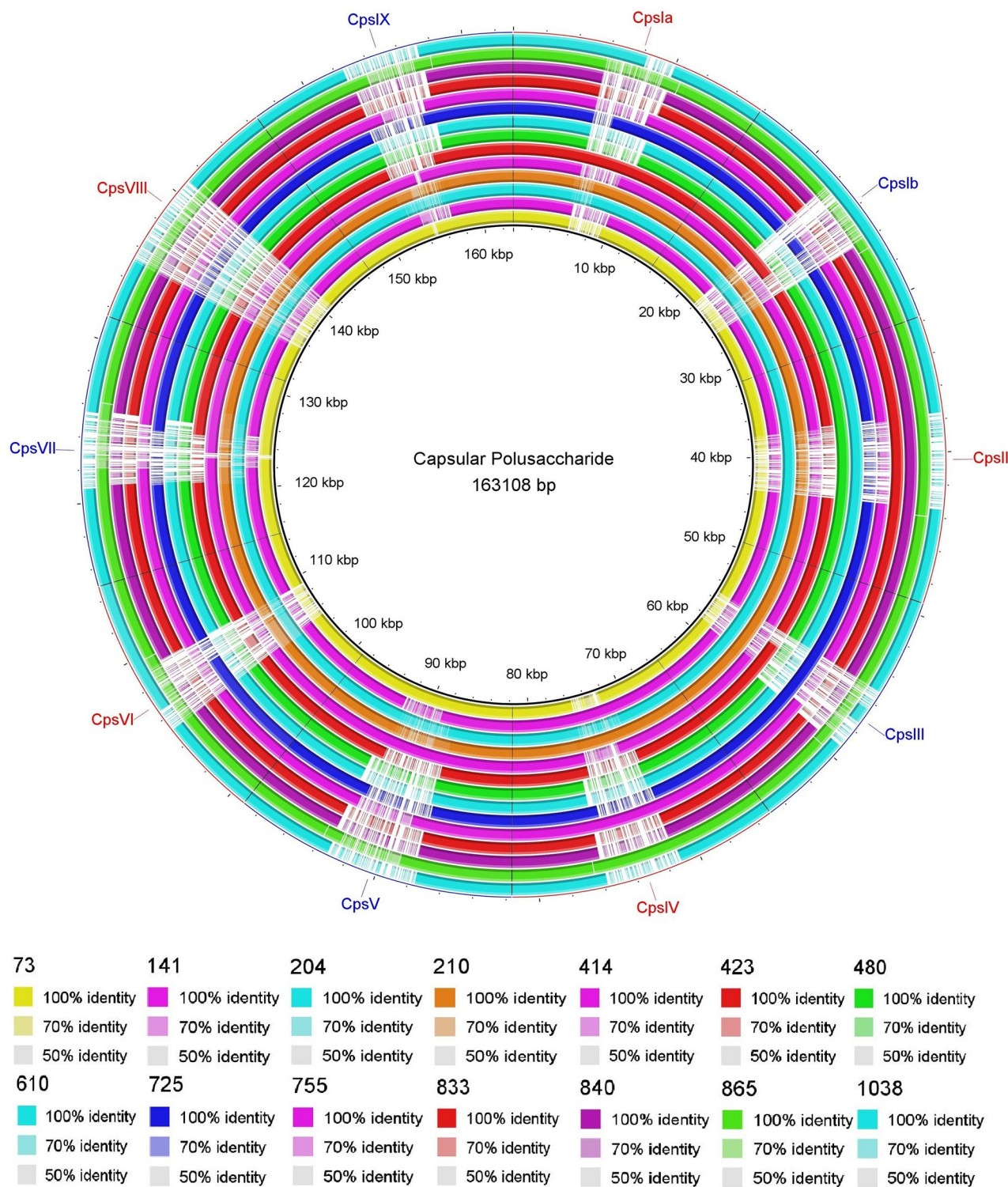

**Fig 1. Circular map of the Cps loci coding for the sialic acid capsular polysaccharide identified in 14 GBS isolates.** Mapping was made by comparative sequence alignment using the BRIG program. Comparative BLASTn analysis with 70, 90, and 100% identities are displayed, and gaps in circles represent regions with no identity of genes.

**Table 7. Virulence gene profiles of 14 group B *Streptococcus* isolated from pregnant women.**

| Virulence Gene Profile | Number of Isolates (%) | Isolates |
| --- | --- | --- |
| *scpB, lmb, PI-1, PI-2a, cylE* | 1 (7.1%) | 141 |
| *scpB, lmb, acp, PI-1, PI-2a, cylE* | 1 (7.1%) | 414 |
| *scpB, lmb, fbsA, acp, PI-2a, cylE* | 1 (7.1%) | 833 |
| *scpB, lmb, fbsA, PI-1, PI-2a, cylE* | 2 (14.3%) | 73, 610 |
| *scpB, lmb, fbsA, acp, PI-1, PI-2a, cylE* | 3 (21.4%) | 480, 840, 1038 |
| *scpB, lmb, fbsA, fbsB, acp, PI-1, PI-2a, cylE* | 1 (7.1%) | 423 |
| *scpB, lmb, fbsA, fbsB, acp, hvgA, PI-2b, cylE* | 1 (7.1%) | 755 |
| *scpB, lmb, fbsA, fbsB, hvgA, PI-1, PI-2b, cylE* | 1 (7.1%) | 725 |
| *scpB, lmb, fbsA, fbsB, acp, hvgA, PI-1, PI-2b, cylE* | 1 (7.1%) | 210 |
| *scpB, lmb, fbsA, fbsB, acp, hvgA, PI-1, PI-2a, PI-2b, cylE* | 2 (14.3%) | 204, 865 |

Mexico showed a higher percentage of sepsis cases and neonatal deaths in children of women who did not receive adequate IAP despite having GBS isolated at the cervicovaginal level or in urine [41]. We did not find transmission of GBS from colonized women to their NB. Moreover, neither mothers nor children had any infectious event attributable to GBS during the three months follow up.

Antibiotic resistance in GBS is of concern due to the role of microorganisms as a leading cause of neonatal disease worldwide [13, 14, 42]. GBS is generally susceptible to beta-lactam antibiotics (including penicillin), the first-line antibiotic for GBS infections and IAP. However, GBS with reduced penicillin susceptibility has been reported more frequently [13]. Resistance to beta-lactam antibiotics in Gram-positive organisms is mainly due to structural changes in the penicillin-binding-proteins (PBPs) caused by acquired mutations in genes that encode PBPs. Instead, resistance to macrolide and lincosamide antibiotics occurs through several mechanisms, including efflux pumps, ribosomal modifications, and drug inactivation. The most widespread resistance mechanism to macrolides is the ribosomal methylation by methyltransferases encoded by *erm* (erythromycin ribosome methylation) genes. In addition, macrolide efflux (Mef) pumps encoded by the *mefA/E* gene are also commonly detected. GBS resistance to tetracycline is attributed to the efflux proteins TetK and TetL or to ribosomal protection proteins TetM and TetO. Resistance to vancomycin results from modifications in the glycopeptide target site through the synthesis of peptidoglycan precursors with altered residues that result in a low affinity for this antibiotic [13, 14, 22]. We assessed antibiotic susceptibility by an automated method or disk diffusion and found that 65% of GBS isolates were resistant to tetracycline. Only 1 of the 17 isolates was sensitive to all antibiotics tested. Previous studies have reported a high rate of resistance to tetracycline, this has been correlated with the presence of the *tetM* and *tetK* genes [43]. Although *tetM* seems to be a determinant of GBS tetracycline resistance, some strains carry a nonfunctional *tetM* conditioning its susceptibility. Even though our isolates are scarce for the proportion of GBS positive to *tetM*, other studies have also reported isolates with this characteristic. There are GBS isolates resistant to tetracycline without the *tetM* and *tetK*, genes, their resistance must be due to other determinants, such as other efflux proteins, ribosomal protection proteins, and enzymatic inactivators [13]. Regarding beta-lactams resistance, we found four strains with reduced susceptibility to beta-lactams. Previous studies have reported that in GBS this characteristic is due to mutations within genes encoding penicillin-biding-proteins (PBP) [13, 44].

Considering that GBS can show high rates of resistance to antibiotics, susceptibility testing of GBS isolated from pregnant women with penicillin allergy is essential for an appropriate drug choice. Furthermore, antibiotic susceptibility testing of all GBS isolated could help to

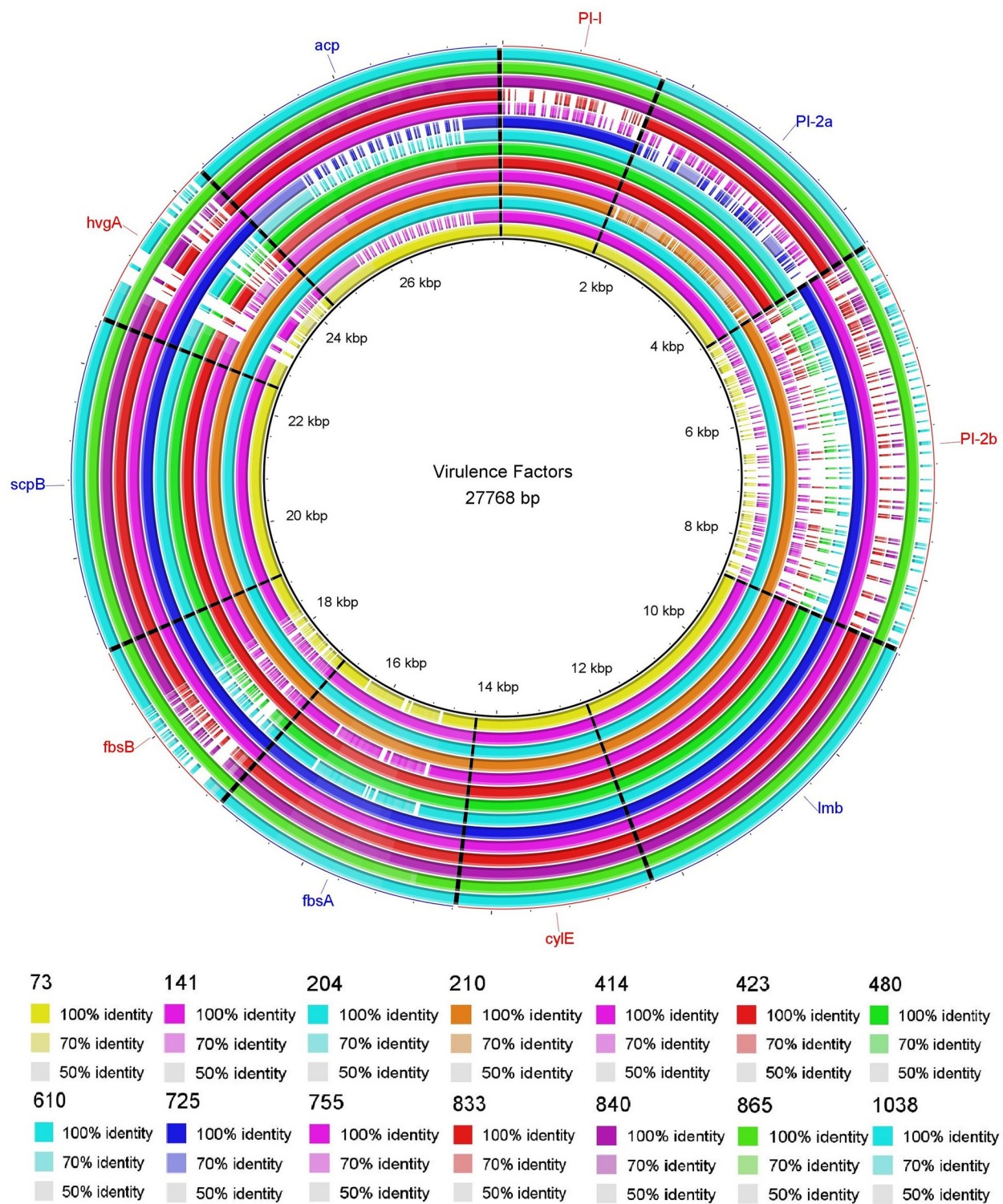

**Fig 2. Localization in a circular map of *scpB*, *lmb*, *fbsA*, *fbsB*, *acp*, *hvgA*, Pili (PI-1, PI-2a and PI-2b), and *cylE* identified in 14 GBS isolates.** Mapping was made by comparative sequence alignment using the BRIG program. Comparative BLASTn analysis with 70, 90 and 100% identities are displayed and gaps in circles represent regions with no identity of genes.

monitor GBS drug resistance [13]. Because IAP is administered to prevent perinatal transmission of GBS, it is also essential to monitor rates of antibiotic resistance. We found a high percentage of resistance in the antimicrobial categories of tetracyclines and penicillins, in addition to non-susceptible isolates to clindamycin and, in one case, to erythromycin. These results are relevant because penicillin is the first-line drug for IAP. Additionally, in cases of a severe allergy to penicillin, clindamycin and erythromycin are recommended as second-line antibiotics in some countries [13, 45]. However, increased resistance to both antibiotics has limited their use. Thus, the Royal College of Obstetricians and Gynecologists (RCOG) no longer recommends the use of clindamycin in the UK, instead recommending vancomycin [46]. Although penicillin remains effective against GBS, increasing reports of isolates with reduced susceptibility are concerning, especially when resistance to second-line antibiotics such as erythromycin and clindamycin remains high among GBS [13, 45].

The comparative genomic analysis of the Cps locus confirmed the serotypes Ia, Ib, II, III, IV, and V previously identified by latex agglutination in our GBS isolates. It is important to notice that the isolates 204 and 865 identified as serotypes III and IV respectively by latex agglutination test showed two Cps each one. Isolate 204 had the Cps II and III and isolate 865 had Cps III and IV. Vagino-rectal co-colonization by multiple GBS serotypes is possible but its detection by standard techniques is complicated, the use of molecular techniques would contribute to a more accurate identification for epidemiological purposes [47].

We identified *lmb*, *scpB*, and *cylE* genes in all isolates, which agree with the study by Udo et al., showing that these three genes are always present in GBS [48]. These genes encode for virulence factors; *lmb* gene encoding for laminin-binding protein, *scpB* encoding for C5a peptidase, which is responsible for cleaving the C5a complement factor and *cylE* encodes the β-hemolysin/cytolysin, a pluripotent toxin associated with the cell surface [49]. The *hvgA* gene encodes an adhesin related with hypervirulence of GBS isolates and it seems to favor not only neonatal colonization but also the development of invasive disease in newborns. The presence of the *hvgA* gene has been reported between 15.1% to 55.6% of the isolates [49, 50]. We found it in 36% of our isolates, but none of them caused disease.

The *acp* gene encoding the surface-associated alpha C protein is responsible for interactions of GBS with epithelial cells and is commonly expressed on the surface of serotypes Ia, Ib, and II [11], we found this gene in most of our isolates, it may be involved in the colonization, but not in causing disease, at least in this study. Fibrinogen-binding protein A (FBPA) encoded by the *fbsA* gene was found in 78.6% of our isolates and detection of the *fbsB* gene encoding for the fibrinogen-binding protein B (FBPB) was found only on *fbsA* positive isolates like Rosenau et al. reported [51]. Two loci code for GBS pili in different regions of the genome designated as pilus islands 1 and 2 (PI-1 and PI-2), the latter presenting two distinct variants, PI-2a and PI-2b [52]. All our GBS isolates had at least one PI gene, the most common combination was PI-1 and PI-2a, but two isolates had all three PI genes. These two isolates were the same having two Cps genes detected. Then, the latter result could be due to co-colonization.

Regarding antibiotic resistance genes, *mprF* (strict) was found in all isolates as previously reported by LaRock and Nizet [53]. This gene provides resistance to defensins and cationic antimicrobial peptides (CAMPs). We also look for *tetM* (strict) and *mefA* (strict) genes involved with tetracyclines resistance. The *tetM* (strict) gene was identified in 78.6% of our isolates, while Zeng X et al. reported, 83% of 512 GBS isolates. The *mefA* (strict) gene was found in only 1 isolated (7%), compared to 22 out of 512 (4%) that Zeng et al. identified. Thus, percentages of *tetM* and *mefA* genes found in this study were similar to those reported by Zeng X et al. [54]. The *mel* gene, involved with macrolide resistance, was identified in 1 of our 14 isolates. This gene has been associated with the presence of *mefA/E* [55]. The isolate 204 (having two Cps) had both genes.

In conclusion, the prevalence of GBS colonization in pregnant women from northeastern Mexico was low (1.47%), and there were no cases of mother-to-child transmission or cases of GBS disease. These results suggest low participation of GBS in perinatal pathology in this population. Consequently, the intentional search for GBS colonization in pregnant women of this region seems not justified, and the indication or not of IAP should be guided only by risk factors. However, follow-up studies with a larger sample number are required to know with certainty the role of GBS in perinatal pathology in this region. Additionally, studies to evaluate if the prevention measures implemented in other countries are also required in Mexico. The analysis of Cps by whole-genome classified the isolates in serotypes Ia, Ib, II, III, IV, or V. These serotypes corresponded to those identified by latex agglutination test, except for two in which two serotypes were found in the same isolate, possibly due to co-colonization by two different GBS. In addition, we detected the genes for virulence factors Lmb, CylE, and ScpB in all our isolates and other virulence genes in variable percentages, among them the major virulence adhesin coded by the *hvgA* gene. These findings suggest that GBS hypervirulent clones are circulating in the population studied, as previously described in a sample of GBS isolates from Mexico City [56]. Although we did not find evidence of GBS disease, it illustrates the importance of knowing the pathogenic characteristics of GBS populations circulating in different regions. On the other hand, we identified resistance or decreased susceptibility to several antibiotics and antimicrobial categories, including the most commonly used antibiotics in IAP, such as penicillins, clindamycin, and erythromycin. This finding shows the importance of adequately knowing the resistance patterns to prevent and treat perinatal GBS infection.

## Author Contributions

**Conceptualization:** Gerardo del Carmen Palacios-Saucedo, Lydia Guadalupe Rivera-Morales, José Manuel Vázquez-Guillén, Ricardo Vázquez-Juárez, Fortino Solórzano-Santos, Cristina Rodríguez-Padilla.

**Data curation:** Melissa Carolina Mellado-García, Aldo Sebastián Flores-Flores, José Alfredo González-Navarro, Celia Geovana Herrera-Rivera, Carolina Barrón-Enríquez, César Alejandro Alonso-Téllez, Luis Daniel Ramírez-Calvillo.

**Formal analysis:** Gerardo del Carmen Palacios-Saucedo, Luis Ernesto Osuna-Rosales, Joaquín Dario Treviño-Baez, César Alejandro Alonso-Téllez, Ricardo Martín Cerda-Flores, Rocío Ortiz-López, Miguel Ángel Rivera-Alvarado, Jorge Castro-Garza.

**Funding acquisition:** Gerardo del Carmen Palacios-Saucedo, Lydia Guadalupe Rivera-Morales.

**Investigation:** Amilcar Caballero-Trejo, Melissa Carolina Mellado-García, Aldo Sebastián Flores-Flores, José Alfredo González-Navarro, Celia Geovana Herrera-Rivera, Luis Ernesto Osuna-Rosales, Julio Antonio Hernández-González, Carolina Barrón-Enríquez, Ramón Valladares-Trujillo, Luis Daniel Ramírez-Calvillo.

**Methodology:** Gerardo del Carmen Palacios-Saucedo, Lydia Guadalupe Rivera-Morales, José Manuel Vázquez-Guillén, Amilcar Caballero-Trejo, Ricardo Vázquez-Juárez, Ricardo Martín Cerda-Flores, Rocío Ortiz-López, Fortino Solórzano-Santos, Cristina Rodríguez-Padilla.

**Project administration:** Gerardo del Carmen Palacios-Saucedo, Lydia Guadalupe Rivera-Morales, José Manuel Vázquez-Guillén.

**Resources:** José Manuel Vázquez-Guillén.

**Supervision:** Gerardo del Carmen Palacios-Saucedo, Lydia Guadalupe Rivera-Morales, José Manuel Vázquez-Guillén, Ricardo Vázquez-Juárez, Ramón Valladares-Trujillo, Joaquín Dario Treviño-Baez, Ricardo Martín Cerda-Flores, Miguel Ángel Rivera-Alvarado.

**Validation:** Jorge Castro-Garza.

**Visualization:** Gerardo del Carmen Palacios-Saucedo, Lydia Guadalupe Rivera-Morales, José Manuel Vázquez-Guillén, Amilcar Caballero-Trejo, Fortino Solórzano-Santos, Cristina Rodríguez-Padilla.

**Writing – original draft:** Gerardo del Carmen Palacios-Saucedo, Lydia Guadalupe Rivera-Morales, José Manuel Vázquez-Guillén, Amilcar Caballero-Trejo, Aldo Sebastián Flores-Flores, José Alfredo González-Navarro, Celia Geovana Herrera-Rivera, Luis Ernesto Osuna-Rosales, Julio Antonio Hernández-González, Ricardo Vázquez-Juárez, Carolina Barrón-Enríquez, Ramón Valladares-Trujillo, Luis Daniel Ramírez-Calvillo.

**Writing – review & editing:** Gerardo del Carmen Palacios-Saucedo, Julio Antonio Hernández-González, Ricardo Vázquez-Juárez, Joaquín Dario Treviño-Baez, César Alejandro Alonso-Téllez, Ricardo Martín Cerda-Flores, Rocío Ortiz-López, Miguel Ángel Rivera-Alvarado, Fortino Solórzano-Santos, Jorge Castro-Garza, Cristina Rodríguez-Padilla.

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
