## [Decision Letter · Decision Letter 0]

15 Oct 2021

PONE-D-21-30812Prevalence, serotype distribution, antimicrobial resistance, and genomic analysis of virulence factors of group B Streptococcus isolated among pregnant women from northeastern Mexico.PLOS ONE

Dear Dr. Rivera-Morales,

Thank you for submitting your manuscript to PLOS ONE. After careful consideration, we feel that it has merit but does not fully meet PLOS ONE’s publication criteria as it currently stands. Therefore, we invite you to submit a revised version of the manuscript that addresses the points raised during the review process.

ACADEMIC EDITOR: Please revise the manuscript according to the reviewer comments==============================

We look forward to receiving your revised manuscript.

Kind regards,

Abdelazeem Mohamed Algammal, Prof, Ph.D

Academic Editor

PLOS ONE

Journal Requirements:

Reviewers' comments:

Reviewer's Responses to Questions

**Comments to the Author**

1. Is the manuscript technically sound, and do the data support the conclusions?

Reviewer #1: Yes

Reviewer #2: Yes

2. Has the statistical analysis been performed appropriately and rigorously? 

Reviewer #1: Yes

Reviewer #2: Yes

3. Have the authors made all data underlying the findings in their manuscript fully available?

Reviewer #1: Yes

Reviewer #2: Yes

4. Is the manuscript presented in an intelligible fashion and written in standard English?

Reviewer #1: Yes

Reviewer #2: Yes

5. Review Comments to the Author

Reviewer #1: Comments to authors:

- The current study is interesting; however, the authors should address the following comments to improve the quality of the manuscript:

- The manuscript should be revised for language editing and grammar mistakes.

- Please write the scientific names of bacterial pathogens in a correct form all over the manuscript and the references section (italic):

For example streptococci, it should be Streptococci (S: should be capital not small, and italic)

Title:

I think the work would benefit from the title that contains the main conclusion of the study (should be derived from the conclusion). Please modify the title.

Abstract:

- The abstract must illustrate the used methods and the most prevalent results (give more hints about methods and results). Besides, rephrase the main conclusion of your findings.

Introduction: (it needs to be more informative)

-Give a hint about different infection caused by Group B Streptococci , virulence factors, and the mechanism of disease occurrence.

- The authors should illustrate the public health importance concerning the emergence of multidrug-resistant (MDR) bacterial pathogens that reflecting the necessity of new potent and safe antimicrobial agents. Several studies proved the widespread MDR- bacterial pathogens;

Authors could add the following paragraph:

Multidrug resistance has been increased all over the world that is considered a public health threat. Several recent investigations reported the emergence of multidrug-resistant bacterial pathogens from different origins including humans, birds, cattle, and fish that increase the need for routine application of the antimicrobial susceptibility testing to detect the antibiotic of choice as well as the screening of the emerging MDR strains. You should cite the following valuable studies:

1.PMID: 33177849

2.PMID: 32497922

3.PMID:33061472

4.PMID: 33947875

5.PMID: 32472209

6.PMID: 31170450

7.PMID: 33188216

8.PMID: 30150182

9. PMID: 34445951

-Rephrase the aim of the work to be clear and better sound.

Material and methods

-Isolation and presumptive

- GBS identification and serotyping identification

•Specific references should be added.

•Add the company, city, and country of the used reagents that were used in the biochemical identification of isolates. Also, enumerate all used biochemical reactions.

- Antimicrobial susceptibility testing:

•Illustrate the antimicrobial classes of the tested antimicrobial agents.

•The authors are advised to classify the tested isolates to MDR or XDR as described by Magiorakos et al.

Magiorakos AP, Srinivasan A, Carey RB, Carmeli Y, Falagas ME, Giske CG, et al. Multidrug-resistant, extensively drug-resistant and pandrug-resistant bacteria: An international expert proposal for interim standard definitions for acquired resistance. Clin Microbiol Infect. 2012; 18:268–81. doi:10.1111/j.1469-0691.2011.03570.x.

-Add more data about the used software in the statistical analyses?

-Result:

-Illustrate the phenotypic characteristics of the recovered GBS isolates.

- Line 220: (Antimicrobial susceptibility) should be modified to be:

Antimicrobial resistance profile of the recovered GBS isolates

-Illustrate in a new table the occurrence of MDR (Multidrug resistance) among the recovered isolates (illustrate the names of the antimicrobial classes and different antibiotics):

No. of strains%Type of resistance

R OR MDR OR XDRPhenotypic multidrug resistance

(Antimicrobial classes and different antibiotics).

The antibiotic -resistance genes

- S1 Fig. and S2 Fig. should be placed in the main manuscript.

-Discussion:

- The authors are advised to illustrate the real impact of their findings without repetition of results.

-Illustrate the different mechanisms of antimicrobial resistance in Streptococci.

-Conclusion

- Should be rephrased to be sounded. A real conclusion should focus on the question or claim you articulated in your study, which resolution has been the main objective of your paper?

Reviewer #2: - The current study has a significant impact, but it needs a minor revision:

- The manuscript should be revised for grammar mistakes.

- Please write the scientific names of bacterial pathogens in the correct form all over the manuscript and in the References section (should be italic).

-The title is broad, please modify the title.

- Add more details about the used methods and most prevalent results in the abstract.

-In the introduction: discuss the public health importance of the Group B Streptococci and different infections caused by them.

-Improve the aim of work.

Methods: Good

-Explain the methods of isolation and identification in detail??

-Specific references should be added to all the used methods and techniques.

-Add the manufacturing company, city, and country for the used reagents and antimicrobial discs.

-Results: Good presentation

- Discuss in detail the phenotypic characters of the isolated GBS strains.

- Please, place the supplementary figures in the main manuscript.

-Discussion:

- Please improve (Avoid repetition of results)

-Please improve the main conclusion of the manuscript.

6. PLOS authors have the option to publish the peer review history of their article (what does this mean?). If published, this will include your full peer review and any attached files.

Reviewer #1: No

Reviewer #2: No

---

## [Author Response · Author response to Decision Letter 0]

3 Feb 2022

Monterrey, Mexico, February 3, 2022. 

Abdelazeem Mohamed Algammal, Prof, Ph.D.

Academic Editor PLOS ONE

Dear Dr. Abdelazeem Mohamed Algammal, Academic Editor of Plos One, this is the rebuttal letter that responds to each point raised by the academic editor and reviewers to the manuscript PONE-D-21-30812: “Genomic analysis of virulence factors and antimicrobial resistance of group B Streptococcus isolated from pregnant women in northeastern Mexico.” We also include a marked-up copy of our manuscript that highlights changes made to the original version and an unmarked version of our revised paper without tracked changes.

Point raised by the academic editor:

• Academic editor comment: If applicable, we recommend that you deposit your laboratory protocols in protocols.io to enhance the reproducibility of your results. Protocols.io assigns your protocol its own identifier (DOI) so that it can be cited independently in the future. For instructions see: https://journals.plos.org/plosone/s/submission-guidelines#loc-laboratory-protocols. Additionally, PLOS ONE offers an option for publishing peer-reviewed Lab Protocol articles, which describe protocols hosted on protocols.io. Read more information on sharing protocols at https://plos.org/protocols?utm_medium=editorial-email&utm_source=authorletters&utm_campaign=protocols.

Change made or response: The laboratory protocols used in our study have been deposited in protocols.io as described below:

Sample collection procedures are available in protocols.io at dx.doi.org/10.17504/protocols.io.b2iaqcae (lines 120-121).

GBS isolation procedure is available in protocols.io at dx.doi.org/10.17504/protocols.io.b2iaqcae (lines 138-139).

GBS identification and serotyping procedures are available in protocols.io at dx.doi.org/10.17504/protocols.io.b2iaqcae, dx.doi.org/10.17504/protocols.io.b2h7qb9n, dx.doi.org/10.17504/protocols.io.b2ibqcan, dx.doi.org/10.17504/protocols.io.b2h8qb9w, and dx.doi.org/10.17504/protocols.io.b2guqbww, dx.doi.org/10.17504/protocols.io.b2uvqew6 (lines 148-152).

• Academic editor comment: Please ensure that your manuscript meets PLOS ONE's style requirements, including those for file naming. The PLOS ONE style templates can be found at 

Change made or response: We have ensured that our manuscript meets the PLOS ONE style requirements, including file naming. For this, we have based ourselves on the PLOS ONE style templates of the two electronic addresses provided.

Reviewers' comments:

Reviewer #1: Comments to authors:

- Reviewer Comment: The current study is interesting; however, the authors should address the following comments to improve the quality of the manuscript: - The manuscript should be revised for language editing and grammar mistakes.

Change made or response: The entire manuscript has been revised in its English edition, and grammatical errors have been removed. 

- Reviewer Comment: Please write the scientific names of bacterial pathogens in a correct form all over the manuscript and the references section (italic): For example streptococci, it should be Streptococci (S: should be capital not small, and italic)

Change made or response: The scientific names of bacterial pathogens were corrected throughout the manuscript and reference section, as indicated by the reviewer. These changes are highlighted in yellow in the marked copy of our manuscript.

- Reviewer Comment: Title: I think the work would benefit from the title that contains the main conclusion of the study (should be derived from the conclusion). Please modify the title.

Change made or response: The title of the work was modified according to the study's main conclusions as follows: “Genomic analysis of virulence factors and antimicrobial resistance of group B Streptococcus isolated from pregnant women in northeastern Mexico”. 

Abstract:

- Reviewer Comment: The abstract must illustrate the used methods and the most prevalent results (give more hints about methods and results). Besides, rephrase the main conclusion of your findings. Introduction: (it needs to be more informative)

Change made or response: In response to the reviewer's recommendations for the abstract, the introduction was made more informative, the methods used were described in more detail, more clues were added about the methods and results of the study, and the main conclusion of the study was rephrased as shown below:

Introduction

The genes lmb, cylE, scpB, and hvgA are involved with increased virulence of GBS, and hypervirulent clones have been identified in different regions. In addition, increasing resistance of GBS to macrolides and lincosamides has been reported, so knowing the patterns of antibiotic resistance may be necessary to prevent and treat GBS infections. This study aimed to identify virulence genes and antibiotic resistance associated with GBS colonization in pregnant women from northeastern Mexico. 

Methods

Pregnant women with 35-37 weeks of gestation underwent recto-vaginal swabbing. One swab was inoculated into Todd-Hewitt broth supplemented with gentamicin and nalidixic acid, a second swab was inoculated into LIM enrichment broth, and a third swab was submerged into a transport medium. All samples were subcultured onto blood agar. After overnight incubation, suggestive colonies with or without hemolysis were analyzed to confirm GBS identification by Gram staining, catalase test, hippurate hydrolysis, CAMP test, and incubation in a chromogenic medium. We used latex agglutination to confirm and serotype GBS isolates. Antibiotic resistance patterns were assessed by Vitek 2 and disk diffusion. Periumbilical, rectal and nasopharyngeal swabs were collected from some newborns of colonized mothers. All colonized women and their newborns were followed up for three months to assess the development of disease attributable to GBS. Draft genomes of all GBS isolates were obtained by whole-genome sequencing. In addition, bioinformatic analysis to identify genes encoding capsular polysaccharides and virulence factors was performed using BRIG, while antibiotic resistance genes were identified using the CARD database.

Results

We found 17 GBS colonized women out of 1154 pregnant women (1.47%). None of the six newborns sampled were colonized, and no complications due to GBS were detected in pregnant women or newborns. Three isolates were serotype I, 5 serotype II, 3 serotype III, 4 serotype IV, and 2 serotype V. Ten distinct virulence gene profiles were identified, being scpB, lmb, fbsA, acp, PI-1, PI-2a, cylE the most common (3/14, 21%). The virulence genes identified were scpB, lmb, cylE, PI-1, fbsA, PI-2a, acp, fbsB, PI-2b, and hvgA. We identified resistance to tetracycline in 65% (11/17) of the isolates, intermediate susceptibility to clindamycin in 41% (7/17), and reduced susceptibility to ampicillin in 23.5% (4/17). The tetM gene associated to tetracyclines resistance was found in 79% (11/14) and the mel and mefA genes associated to macrolides resistance in 7% (1/14). 

Conclusions

The detection of strains with genes coding virulence factors means that clones with pathogenic potential circulates in this region. On the other hand, the identification of decreased susceptibility to antibiotics from different antimicrobial categories shows the importance of adequately knowing the resistance patterns to prevent and to treat GBS perinatal infection.

- Reviewer Comment: Give a hint about different infection caused by Group B Streptococci, virulence factors, and the mechanism of disease occurrence.

Change made or response: As directed by the reviewer, more information about the different types of GBS infections, their virulence factors, and the mechanisms of disease occurrence have been added to the introduction as shown below:

Group B Streptococcus (GBS; Streptococcus agalactiae) colonizes the human genitourinary and gastrointestinal tract and causes a variety of infectious processes in pregnant women, such as asymptomatic bacteriuria, urinary tract infection, bacteremia, pneumonia, meningitis, endocarditis, sepsis, and various obstetric complications as spontaneous abortion, premature labor, chorioamnionitis, endometritis, stillbirth, and neonatal and maternal death. In addition, according to the onset of clinical manifestations, GBS infection in newborns (NB) can present with bacteremia, pneumonia, and sepsis as early-onset disease, late-onset disease, or late-late-onset disease [1, 2]. (Line 81-88) 

Previous studies reported several factors making GBS more virulent and resistant to antibiotics. The most studied virulence factor is the capsular polysaccharide (Cps), which defines GBS serotypes (Ia, Ib, II-IX) and contributes to evade the immune system. However, other factors, such as laminin-binding protein (Lmb), fibrinogens (Fbs), hypervirulent adhesin (HvgA), and alpha-C protein (ACP), are associated with adherence and cell invasion [9, 10, 11]. In addition, an increasing GBS resistance to macrolides, lincosamides, and tetracyclines has been reported by several authors worldwide [12, 13]. (Line 98-104) 

- Reviewer Comment: The authors should illustrate the public health importance concerning the emergence of multidrug-resistant (MDR) bacterial pathogens that reflecting the necessity of new potent and safe antimicrobial agents. Several studies proved the widespread MDR- bacterial pathogens;

Authors could add the following paragraph:

Multidrug resistance has been increased all over the world that is considered a public health threat. Several recent investigations reported the emergence of multidrug-resistant bacterial pathogens from different origins including humans, birds, cattle, and fish that increase the need for routine application of the antimicrobial susceptibility testing to detect the antibiotic of choice as well as the screening of the emerging MDR strains. You should cite the following valuable studies:

1.PMID: 33177849

2.PMID: 32497922

3.PMID:33061472

4.PMID: 33947875

5.PMID: 32472209

6.PMID: 31170450

7.PMID: 33188216

8.PMID: 30150182

9.PMID: 34445951

Change made or response: In response to this reviewer's recommendation, we added the following statement at the end of the introduction and included several of the references suggested by this reviewer:

Multidrug resistance has been increased all over the world that is considered a public health threat. Several recent investigations reported the emergence of multidrug-resistant bacterial pathogens from different origins, including humans, birds, cattle, and fish, that increase the need for routine application of antimicrobial susceptibility testing to detect the antibiotic of choice and the screening of the emerging MDR strains [13-17]. (Line 104-108)

- Reviewer Comment: Rephrase the aim of the work to be clear and better sound.

Change made or response: The aim of the study was rephrased as follows:

The aim of the present study was to explore the presence of virulence and antibiotic resistance genes in GBS associated with colonization in pregnant women in a population from northeastern Mexico. (Line 108-110)

Material and methods

- Reviewer Comment: Isolation and presumptive GBS identification and serotyping identification

•Specific references should be added.

Change made or response: The laboratory protocols used in our study were deposited in protocols.io as described below:

Sample collection procedures are available in protocols.io at dx.doi.org/10.17504/protocols.io.b2iaqcae (Line 120-121).

GBS isolation procedure used is available in protocols.io at dx.doi.org/10.17504/protocols.io.b2iaqcae (Line 138-139).

GBS identification and serotyping procedures are available in protocols.io at dx.doi.org/10.17504/protocols.io.b2iaqcae, dx.doi.org/10.17504/protocols.io.b2h7qb9n, dx.doi.org/10.17504/protocols.io.b2ibqcan, dx.doi.org/10.17504/protocols.io.b2h8qb9w, dx.doi.org/10.17504/protocols.io.b2guqbww, and dx.doi.org/10.17504/protocols.io.b2uvqew6 (Line 148-152).

In addition, we added the following references to support the isolation, identification, and serotyping methods used:

19. Filkins L, Hauser J, Robinson-Dunn B, Tibbetts R, Boyanton B, Revell P, on behalf of the American Society for Microbiology Clinical and Public Health Microbiology Committee, Subcommittee on Laboratory Practices. 2020. Guidelines for detection and identification of group B Streptococcus. American Society for Microbiology. https://asm.org/Guideline/Guidelines-for-the-Detection-and-Identification-of.

20. Hansen SM, Sørensen UB. Method for quantitative detection and presumptive identification of group B Streptococci on primary plating. J Clin Microbiol. 2003 Apr;41(4):1399-403. doi: 10.1128/JCM.41.4.1399-1403.2003.

21. Slotved HC, Elliott J, Thompson T, Konradsen HB. Latex assay for serotyping of group B Streptococcus isolates. J Clin Microbiol. 2003;41(9):4445-7. doi: 10.1128

- Reviewer Comment: Add the company, city, and country of the used reagents that were used in the biochemical identification of isolates. Also, enumerate all used biochemical reactions.

Change made or response: We added the company, city, and country of all reagents used in the biochemical identification of the isolates, and all the biochemical reactions used were listed. (Lines 132, 170, 172, 175, and 176)

- Reviewer Comment: Antimicrobial susceptibility testing:

• Illustrate the antimicrobial classes of the tested antimicrobial agents.

•The authors are advised to classify the tested isolates to MDR or XDR as described by Magiorakos et al.

Magiorakos AP, Srinivasan A, Carey RB, Carmeli Y, Falagas ME, Giske CG, et al. Multidrug-resistant, extensively drug-resistant and pandrug-resistant bacteria: An international expert proposal for interim standard definitions for acquired resistance. Clin Microbiol Infect. 2012; 18:268–81. doi:10.1111/j.1469-0691.2011.03570.x. 

Change made or response: The different classes of antimicrobials tested were added as a list, as shown below:

The categories of antimicrobials tested were fluoroquinolones, glycopeptides, glycylcyclines, oxazolidinones, penicillins, tetracyclines, in addition to erythromycin and clindamycin [22]. (Lines 155 and 156) 

We also added the classification of GBS isolates based on the definition by Magiorakos et al. in the results section.

- Reviewer Comment: Add more data about the used software in the statistical analyses?

Change made or response Change made or response: We completed the information of the software used in the statistical analysis as shown below:

The analysis was made using the IBM SPSS Statistics software (Version 20; SPSS Inc., Armonk, NY, USA). (Lines 196 and 197)

- Result:

- Reviewer Comment: Illustrate the phenotypic characteristics of the recovered GBS isolates.

- Change made or response: 

Phenotypic characteristics of GBS isolates are shown in different parts of the text and tables in the results section and include serotype and antimicrobial resistance profiles. (Lines 232-279) 

The genotypic characteristics from the genome-wide analysis are also described in the text, tables, and figures. (Lines 281-337)

- Reviewer Comment: Line 220: (Antimicrobial susceptibility) should be modified to be: Antimicrobial resistance profile of the recovered GBS isolates

Change made or response: Based on this recommendation, we changed the subtitle of this section to read “Antimicrobial resistance profile”. (Line 246) 

- Reviewer Comment: Illustrate in a new table the occurrence of MDR (Multidrug resistance) among the recovered isolates (illustrate the names of the antimicrobial classes and different antibiotics):

No. of strains%Type of resistance

R OR MDR OR XDRPhenotypic multidrug resistance

(Antimicrobial classes and different antibiotics).

The antibiotic -resistance genes

Change made or response: In this new version of our manuscript, the information on antimicrobial resistance is presented in four tables.

• Table 3 describes the level of antibiotic resistance and serotype of GBS isolates. (Lines 255-258)

• Table 4 presents the antimicrobial susceptibility, along with the number of antimicrobials and the number of antimicrobial categories to which the isolates showed resistance or decreased susceptibility according to the classification by Magiorakos et al. (Lines 268-273)

• Table 5 shows the antimicrobial resistance of GBS isolates classified by agents and antimicrobial categories according to Magiorakos et al. (Lines 276-279)

• The information on identifying genes related to antimicrobial resistance is included in Table 6. (Lines 324-333). 

- Reviewer Comment: S1 Fig. and S2 Fig. should be placed in the main manuscript.

Change made or response: Figure 1 and Figure 2 are part of the main manuscript in this new version. (Lines 292-296 and 312-316)

- Discussion:

- Reviewer Comment: The authors are advised to illustrate the real impact of their findings without repetition of results.

Change made or response: To illustrate the real impact of our findings, we added the following paragraphs in the discussion section: 

• Because IAP is administered to prevent perinatal transmission of GBS, it is also essential to monitor rates of antibiotic resistance. We found a high percentage of resistance in the antimicrobial categories of tetracyclines and penicillins, in addition to non-susceptible isolates to clindamycin and, in one case, to erythromycin. These results are relevant because penicillin is the first-line drug for IAP. Additionally, in cases of a severe allergy to penicillin, clindamycin and erythromycin are recommended as second-line antibiotics in some countries [13, 45]. However, increased resistance to both antibiotics has limited their use. Thus, the Royal College of Obstetricians and Gynecologists (RCOG) no longer recommends the use of clindamycin in the UK, instead recommending vancomycin [46]. Although penicillin remains effective against GBS, increasing reports of isolates with reduced susceptibility are concerning, especially when resistance to second-line antibiotics such as erythromycin and clindamycin remains high among GBS [13, 45]. (Lines 410-422)

• In addition, we detected the genes for virulence factors Lmb, CylE, and ScpB in all our isolates and other virulence genes in variable percentages, among them the major virulence adhesin coded by the hvgA gene. These findings suggest that GBS hypervirulent clones are circulating in the population studied, as previously described in a sample of GBS isolates from Mexico City [56]. Although we did not find evidence of GBS disease, they illustrate the importance of knowing the pathogenic characteristics of GBS populations circulating in different regions. On the other hand, we identified resistance or decreased susceptibility to several antibiotics and antimicrobial categories, including the most commonly used antibiotics in IAP, such as penicillins, clindamycin, and erythromycin. This finding shows the importance of adequately knowing the resistance patterns to prevent and treat perinatal GBS infection. (Lines 473-483)

- Reviewer Comment: Illustrate the different mechanisms of antimicrobial resistance in Streptococci.

Change made or response: To illustrate the different mechanisms of antimicrobial resistance in GBS, we also add the following paragraph: 

• Antibiotic resistance in GBS is of concern due to the role of microorganisms as a leading cause of neonatal disease worldwide [13, 14, 42]. GBS is generally susceptible to beta-lactam antibiotics (including penicillin), the first-line antibiotic for GBS infections and IAP. However, GBS with reduced penicillin susceptibility has been reported more frequently [13]. Resistance to beta-lactam antibiotics in Gram-positive organisms is mainly due to structural changes in the penicillin-binding-proteins (PBPs) caused by acquired mutations in genes that encode PBPs. Instead, resistance to macrolide and lincosamide antibiotics occurs through several mechanisms, including efflux pumps, ribosomal modifications, and drug inactivation. The most widespread resistance mechanism to macrolides is the ribosomal methylation by methyltransferases encoded by erm (erythromycin ribosome methylation) genes. In addition, macrolide efflux (Mef) pumps encoded by the mefA/E gene are also commonly detected. GBS resistance to tetracycline is attributed to the efflux proteins TetK and TetL or to ribosomal protection proteins TetM and TetO. Resistance to vancomycin results from modifications in the glycopeptide target site through the synthesis of peptidoglycan precursors with altered residues that result in a low affinity for this antibiotic [13, 14, 22]. (Lines 378-393)

- Conclusion

- Reviewer Comment: Should be rephrased to be sounded. A real conclusion should focus on the question or claim you articulated in your study, which resolution has been the main objective of your paper?

Change made or response: To focus the conclusion on the objective and main findings of the study, we added the following information in the conclusion paragraph:

• In addition, we detected the genes for virulence factors Lmb, CylE, and ScpB in all our isolates and other virulence genes in variable percentages, among them the major virulence adhesin coded by the hvgA gene. These findings suggest that GBS hypervirulent clones are circulating in the population studied, as previously described in a sample of GBS isolates from Mexico City [56]. Although we did not find evidence of GBS disease, they illustrate the importance of knowing the pathogenic characteristics of GBS populations circulating in different regions. On the other hand, we identified resistance or decreased susceptibility to several antibiotics and antimicrobial categories, including the most commonly used antibiotics in IAP, such as penicillins, clindamycin, and erythromycin. This finding shows the importance of adequately knowing the resistance patterns to prevent and treat perinatal GBS infection. (Lines 473-483)

Reviewer #2: The current study has a significant impact, but it needs a minor revision:

- Reviewer Comment: The manuscript should be revised for grammar mistakes.

Change made or response: The whole manuscript has been revised in its English edition, and grammatical errors have been removed. 

- Reviewer Comment: Please write the scientific names of bacterial pathogens in the correct form all over the manuscript and in the References section (should be italic).

Change made or response: The scientific names of bacterial pathogens were corrected throughout the manuscript and reference section, as indicated by the reviewer. These changes are highlighted in yellow in the marked copy of our manuscript.

- Reviewer Comment: The title is broad, please modify the title.

Change made or response: The work title was modified according to the study's main conclusions as follows: “Genomic analysis of virulence factors and antimicrobial resistance of group B Streptococcus isolated from pregnant women in northeastern Mexico”. 

- Reviewer Comment: Add more details about the used methods and most prevalent results in the abstract.

Change made or response: Within the abstract introduction was made more informative, the methods used were described in more detail, more clues were added about the methods and results, and the main conclusion of the study was rephrased as shown below:

Introduction

The genes lmb, cylE, scpB, and hvgA are involved with increased virulence of GBS, and hypervirulent clones have been identified in different regions. In addition, increasing resistance of GBS to macrolides and lincosamides has been reported, so knowing the patterns of antibiotic resistance may be necessary to prevent and treat GBS infections. This study aimed to identify virulence genes and antibiotic resistance associated with GBS colonization in pregnant women from northeastern Mexico. 

Methods

Pregnant women with 35-37 weeks of gestation underwent recto-vaginal swabbing. One swab was inoculated into Todd-Hewitt broth supplemented with gentamicin and nalidixic acid, a second swab was inoculated into LIM enrichment broth, and a third swab was submerged into a transport medium. All samples were subcultured onto blood agar. After overnight incubation, suggestive colonies with or without hemolysis were analyzed to confirm GBS identification by Gram staining, catalase test, hippurate hydrolysis, CAMP test, and incubation in a chromogenic medium. We used latex agglutination to confirm and serotype GBS isolates. Antibiotic resistance patterns were assessed by Vitek 2 and disk diffusion. Periumbilical, rectal and nasopharyngeal swabs were collected from some newborns of colonized mothers. All colonized women and their newborns were followed up for three months to assess the development of disease attributable to GBS. Draft genomes of all GBS isolates were obtained by whole-genome sequencing. In addition, bioinformatic analysis to identify genes encoding capsular polysaccharides and virulence factors was performed using BRIG, while antibiotic resistance genes were identified using the CARD database.

Results

We found 17 GBS colonized women out of 1154 pregnant women (1.47%). None of the six newborns sampled were colonized, and no complications due to GBS were detected in pregnant women or newborns. Three isolates were serotype I, 5 serotype II, 3 serotype III, 4 serotype IV, and 2 serotype V. Ten distinct virulence gene profiles were identified, being scpB, lmb, fbsA, acp, PI-1, PI-2a, cylE the most common (3/14, 21%). The virulence genes identified were scpB, lmb, cylE, PI-1, fbsA, PI-2a, acp, fbsB, PI-2b, and hvgA. We identified resistance to tetracycline in 65% (11/17) of the isolates, intermediate susceptibility to clindamycin in 41% (7/17), and reduced susceptibility to ampicillin in 23.5% (4/17). The tetM gene associated to tetracyclines resistance was found in 79% (11/14) and the mel and mefA genes associated to macrolides resistance in 7% (1/14). 

Conclusions

The detection of strains with genes coding virulence factors means that clones with pathogenic potential circulates in this region. On the other hand, the identification of decreased susceptibility to antibiotics from different antimicrobial categories shows the importance of adequately knowing the resistance patterns to prevent and to treat GBS perinatal infection.

- Reviewer Comment: In the introduction: discuss the public health importance of the Group B Streptococci and different infections caused by them.

Change made or response: We added more information in the introduction section about the different types of GBS infections, virulence factors, and mechanisms of disease as shown below:

• Group B Streptococcus (GBS; Streptococcus agalactiae) colonizes the human genitourinary and gastrointestinal tract and causes a variety of infectious processes in pregnant women, such as asymptomatic bacteriuria, urinary tract infection, bacteremia, pneumonia, meningitis, endocarditis, sepsis, and various obstetric complications as spontaneous abortion, premature labor, chorioamnionitis, endometritis, stillbirth, and neonatal and maternal death. In addition, according to the onset of clinical manifestations, GBS infection in newborns (NB) can present with bacteremia, pneumonia, and sepsis as early-onset disease, late-onset disease, or late-late-onset disease [1, 2]. (Line 81-88) 

• Previous studies reported several factors making GBS more virulent and resistant to antibiotics. The most studied virulence factor is the capsular polysaccharide (Cps), which defines GBS serotypes (Ia, Ib, II-IX) and contributes to evade the immune system. However, other factors, such as laminin-binding protein (Lmb), fibrinogens (Fbs), hypervirulent adhesin (HcgA), and alpha-C protein (ACP), are associated with adherence and cell invasion [9, 10, 11]. In addition, an increasing GBS resistance to macrolides, lincosamides, and tetracyclines has been reported by several authors worldwide [12, 13]. (Line 98-104) 

• Multidrug resistance has been increased all over the world that is considered a public health threat. Several recent investigations reported the emergence of multidrug-resistant bacterial pathogens from different origins, including humans, birds, cattle, and fish, that increase the need for routine application of antimicrobial susceptibility testing to detect the antibiotic of choice and the screening of the emerging MDR strains [13-17]. (Line 104-108)

- Reviewer Comment: Improve the aim of work.

Change made or response: The aim of the study was rephrased as follows:

The aim of the present study was to explore the presence of virulence and antibiotic resistance genes in GBS associated with colonization in pregnant women in a population from northeastern Mexico. (Line 108-110)

Methods: Good

- Reviewer Comment: Explain the methods of isolation and identification in detail??

Change made or response: The laboratory protocols used in our study were deposited and are available in protocols.io as described below:

Sample collection procedures are available in protocols.io at dx.doi.org/10.17504/protocols.io.b2iaqcae (Line 120-121).

GBS isolation procedure used is available in protocols.io at dx.doi.org/10.17504/protocols.io.b2iaqcae (Line 138-139).

GBS identification and serotyping procedures are available in protocols.io at dx.doi.org/10.17504/protocols.io.b2iaqcae, dx.doi.org/10.17504/protocols.io.b2h7qb9n, dx.doi.org/10.17504/protocols.io.b2ibqcan, dx.doi.org/10.17504/protocols.io.b2h8qb9w, dx.doi.org/10.17504/protocols.io.b2guqbww, and dx.doi.org/10.17504/protocols.io.b2uvqew6 (Line 148-152).

The different classes of antimicrobials tested were added as a list, as follows: 

The categories of antimicrobials tested were fluoroquinolones, glycopeptides, glycylcyclines, oxazolidinones, penicillins, tetracyclines, in addition to erythromycin and clindamycin [22]. (Lines 154 and 156)

Additionally, we added the classification of GBS isolates based on the definition by Magiorakos et al. in the results section. 

We also completed the information of the software used in the statistical analysis as follows: The analysis was made using the IBM SPSS Statistics software (Version 20; SPSS Inc., Armonk, NY, USA). (Lines 196 and 197)

- Specific references should be added to all the used methods and techniques.

Change made or response: We added the following references to support the isolation, identification, and serotyping methods used:

19. Filkins L, Hauser J, Robinson-Dunn B, Tibbetts R, Boyanton B, Revell P, on behalf of the American Society for Microbiology Clinical and Public Health Microbiology Committee, Subcommittee on Laboratory Practices. 2020. Guidelines for detection and identification of group B Streptococcus. American Society for Microbiology. https://asm.org/Guideline/Guidelines-for-the-Detection-and-Identification-of.

20. Hansen SM, Sørensen UB. Method for quantitative detection and presumptive identification of group B streptococci on primary plating. J Clin Microbiol. 2003 Apr;41(4):1399-403. doi: 10.1128/JCM.41.4.1399-1403.2003.

21. Slotved HC, Elliott J, Thompson T, Konradsen HB. Latex assay for serotyping of group B Streptococcus isolates. J Clin Microbiol. 2003;41(9):4445-7. doi: 10.1128

- Reviewer Comment: Add the manufacturing company, city, and country for the used reagents and antimicrobial discs.

Change made or response: We added the company, city, and country of all reagents used in the biochemical identification of the isolates, and all the biochemical reactions used were listed. (Lines 132, 170, 172, 175, and 176)

- Reviewer Comment: Results: Good presentation

Change made or response: Thank you very much for your comment.

- Reviewer Comment: Discuss in detail the phenotypic characters of the isolated GBS strains.

Change made or response: 

Phenotypic characteristics of GBS isolates are shown in different parts of the text and tables in the results section and include serotype and antimicrobial resistance profiles. (Lines 232-279) 

The genotypic characteristics from the genome-wide analysis are also described in the text, tables, and figures. (Lines 281-337)

- Reviewer Comment: Please, place the supplementary figures in the main manuscript.

Change made or response: Figures 1 and Figure 2 are part of the main manuscript in this new version. (Lines 292-296 and 312-316)

- Discussion:

- Reviewer Comment: Please improve (Avoid repetition of results)

Change made or response: We added the following paragraphs discussing our phenotypic and genotypic findings related to antibiotic resistance: 

• Because IAP is administered to prevent perinatal transmission of GBS, it is also essential to monitor rates of antibiotic resistance. We found a high percentage of resistance in the antimicrobial categories of tetracyclines and penicillins, in addition to non-susceptible isolates to clindamycin and, in one case, to erythromycin. These results are relevant because penicillin is the first-line drug for IAP. Additionally, in cases of a severe allergy to penicillin, clindamycin and erythromycin are recommended as second-line antibiotics in some countries [13, 45]. However, increased resistance to both antibiotics has limited their use. Thus, the Royal College of Obstetricians and Gynecologists (RCOG) no longer recommends the use of clindamycin in the UK, instead recommending vancomycin [46]. Although penicillin remains effective against GBS, increasing reports of isolates with reduced susceptibility are concerning, especially when resistance to second-line antibiotics such as erythromycin and clindamycin remains high among GBS [13, 45]. (Lines 410-422)

• In addition, we detected the genes for virulence factors Lmb, CylE, and ScpB in all our isolates and other virulence genes in variable percentages, among them the major virulence adhesin coded by the hvgA gene. These findings suggest that GBS hypervirulent clones are circulating in the population studied, as previously described in a sample of GBS isolates from Mexico City [56]. Although we did not find evidence of GBS disease, they illustrate the importance of knowing the pathogenic characteristics of GBS populations circulating in different regions. On the other hand, we identified resistance or decreased susceptibility to several antibiotics and antimicrobial categories, including the most commonly used antibiotics in IAP, such as penicillins, clindamycin, and erythromycin. This finding shows the importance of adequately knowing the resistance patterns to prevent and treat perinatal GBS infection. (Lines 473-483)

• Antibiotic resistance in GBS is of concern due to the role of microorganisms as a leading cause of neonatal disease worldwide [13, 14, 42]. GBS is generally susceptible to beta-lactam antibiotics (including penicillin), the first-line antibiotic for GBS infections and IAP. However, GBS with reduced penicillin susceptibility has been reported more frequently [13]. Resistance to beta-lactam antibiotics in Gram-positive organisms is mainly due to structural changes in the penicillin-binding-proteins (PBPs) caused by acquired mutations in genes that encode PBPs. Instead, resistance to macrolide and lincosamide antibiotics occurs through several mechanisms, including efflux pumps, ribosomal modifications, and drug inactivation. The most widespread resistance mechanism to macrolides is the ribosomal methylation by methyltransferases encoded by erm (erythromycin ribosome methylation) genes. In addition, macrolide efflux (Mef) pumps encoded by the mefA/E gene are also commonly detected. GBS resistance to tetracycline is attributed to the efflux proteins TetK and TetL or to ribosomal protection proteins TetM and TetO. Resistance to vancomycin results from modifications in the glycopeptide target site through the synthesis of peptidoglycan precursors with altered residues that result in a low affinity for this antibiotic [13, 14, 22]. (Lines 378-393)

- Reviewer Comment: Please improve the main conclusion of the manuscript.

Change made or response: To focus the conclusion on the objective and main findings of the study, we added the following information in the conclusion paragraph:

• In addition, we detected the genes for virulence factors Lmb, CylE, and ScpB in all our isolates and other virulence genes in variable percentages, among them the major virulence adhesin coded by the hvgA gene. These findings suggest that GBS hypervirulent clones are circulating in the population studied, as previously described in a sample of GBS isolates from Mexico City [56]. Although we did not find evidence of GBS disease, they illustrate the importance of knowing the pathogenic characteristics of GBS populations circulating in different regions. On the other hand, we identified resistance or decreased susceptibility to several antibiotics and antimicrobial categories, including the most commonly used antibiotics in IAP, such as penicillins, clindamycin, and erythromycin. This finding shows the importance of adequately knowing the resistance patterns to prevent and treat perinatal GBS infection. (Lines 473-483)

6. PLOS authors have the option to publish the peer review history of their article (what does this mean?). If published, this will include your full peer review and any attached files.

Response: We agree that the peer review history of our article is published, including the full peer review and any attached files.

We have considered all the suggestions made by the reviewers, and the changes made are highlighted in yellow within the text of one of the files we uploaded. We hope that the modifications introduced may be appropriate and that our manuscript may be considered for publication in Plos One, and we look forward to a prompt response. Many thanks for all your attention. 

Dr. Gerardo del Carmen Palacios Saucedo

Main author

---

## [Decision Letter · Decision Letter 1]

8 Feb 2022

Genomic analysis of virulence factors and antimicrobial resistance of group B Streptococcus isolated from pregnant women in northeastern Mexico

PONE-D-21-30812R1

Dear Dr. Rivera-Morales,

We’re pleased to inform you that your manuscript has been judged scientifically suitable for publication and will be formally accepted for publication once it meets all outstanding technical requirements.

Kind regards,

Abdelazeem Mohamed Algammal, Prof, Ph.D

Academic Editor

PLOS ONE

Additional Editor Comments (optional):

Reviewers' comments:

Reviewer's Responses to Questions

**Comments to the Author**

1. If the authors have adequately addressed your comments raised in a previous round of review and you feel that this manuscript is now acceptable for publication, you may indicate that here to bypass the “Comments to the Author” section, enter your conflict of interest statement in the “Confidential to Editor” section, and submit your "Accept" recommendation.

Reviewer #1: All comments have been addressed

2. Is the manuscript technically sound, and do the data support the conclusions?

Reviewer #1: Yes

3. Has the statistical analysis been performed appropriately and rigorously? 

Reviewer #1: Yes

4. Have the authors made all data underlying the findings in their manuscript fully available?

Reviewer #1: Yes

5. Is the manuscript presented in an intelligible fashion and written in standard English?

Reviewer #1: Yes

6. Review Comments to the Author

Reviewer #1: The authors have carried out significant changes to the manuscript. They have addressed all the suggested corrections and comments. Really, it's an interesting study that has a significant impact. Now, the manuscript could be accepted.

7. PLOS authors have the option to publish the peer review history of their article (what does this mean?). If published, this will include your full peer review and any attached files.

Reviewer #1: No

---

## [Editor Report · Acceptance letter]

5 Mar 2022

PONE-D-21-30812R1 

Genomic analysis of virulence factors and antimicrobial resistance of group B *Streptococcus* isolated from pregnant women in northeastern Mexico 

Dear Dr. Rivera-Morales:

I'm pleased to inform you that your manuscript has been deemed suitable for publication in PLOS ONE. Congratulations! Your manuscript is now with our production department. 

Kind regards, 

on behalf of

Professor Abdelazeem Mohamed Algammal 

Academic Editor

PLOS ONE